# Mature Andean forests as globally important carbon sinks and future carbon refuges

Alvaro Duque [1✉], Miguel A. Peña[1], Francisco Cuesta[2], Sebastián González-Caro[1], Peter Kennedy[3], Oliver L. Phillips [4], Marco Calderón-Loor [2,5], Cecilia Blundo [6], Julieta Carilla [6], Leslie Cayola[7,8], William Farfán-Ríos [9,10], Alfredo Fuentes[7,8], Ricardo Grau[6], Jürgen Homeier [11,12], María I. Loza-Rivera[7,8,9], Yadvinder Malhi [13], Agustina Malizia [6], Lucio Malizia[14], Johanna A. Martínez-Villa [15], Jonathan A. Myers [16], Oriana Osinaga-Acosta[6], Manuel Peralvo[17], Esteban Pinto[2,18], Sassan Saatchi[19], Miles Silman[20], J. Sebastián Tello[9], Andrea Terán-Valdez[21] & Kenneth J. Feeley [22]

It is largely unknown how South America's Andean forests affect the global carbon cycle, and thus regulate climate change. Here, we measure aboveground carbon dynamics over the past two decades in 119 monitoring plots spanning a range of >3000 m elevation across the subtropical and tropical Andes. Our results show that Andean forests act as strong sinks for aboveground carbon ($0.67 \pm 0.08$ Mg C ha$^{-1}$ y$^{-1}$) and have a high potential to serve as future carbon refuges. Aboveground carbon dynamics of Andean forests are driven by abiotic and biotic factors, such as climate and size-dependent mortality of trees. The increasing aboveground carbon stocks offset the estimated C emissions due to deforestation between 2003 and 2014, resulting in a net total uptake of 0.027 Pg C y$^{-1}$. Reducing deforestation will increase Andean aboveground carbon stocks, facilitate upward species migrations, and allow for recovery of biomass losses due to climate change.

[1] Departamento de Ciencias Forestales, Universidad Nacional de Colombia Sede Medellín, Medellín, Colombia. [2] Grupo de Investigación en Biodiversidad, Medio Ambiente y Salud -BIOMAS - Universidad de Las Américas (UDLA), Quito, Ecuador. [3] Department of Plant and Microbial Biology, University of Minnesota, Saint Paul, MN, USA. [4] School of Geography, University of Leeds, Leeds, UK. [5] Centre for Integrative Ecology, School of Life and Environmental Sciences, Deakin University, Melbourne, VIC, Australia. [6] Instituto de Ecología Regional (IER), Universidad Nacional de Tucumán (UNT) - Consejo Nacional de Investigaciones Científicas y Técnicas (CONICET), Tucumán, Argentina. [7] Herbario Nacional de Bolivia (LPB), La Paz, Bolivia. [8] Missouri Botanical Garden, St. Louis, MO, USA. [9] Center for Conservation and Sustainable Development, Missouri Botanical Garden, St. Louis, MO, USA. [10] Living Earth Collaborative, Washington University in Saint Louis, St. Louis, MO, USA. [11] Plant Ecology and Ecosystems Research, University of Gottingen, Gottingen, Germany. [12] Centre of Biodiversity and Sustainable Land Use (CBL), University of Gottingen, Gottingen, Germany. [13] Environmental Change Institute, School of Geography and the Environment, University of Oxford, Oxford, UK. [14] Facultad de Ciencias Agrarias, Universidad Nacional de Jujuy, Jujuy, Argentina. [15] Université du Quebec a Montreal, Montreal, QC, Canada. [16] Department of Biology, Washington University in St. Louis, St. Louis, MO, USA. [17] Consorcio para el Desarrollo Sostenible de la Ecorregión Andina (CONDESAN), Quito, Ecuador. [18] Columbus State University, University System of Georgia, Columbus, GA, USA. [19] Carbon Cycle and Ecosystems, Jet Propulsion Laboratory, California Institute of Technology, Pasadena, CA, USA. [20] Center for Energy, Environment and Sustainability, Winston-Salem, NC, USA. [21] Centro Jambatú de Investigación y Conservación de Anfibios, Quito, Ecuador. [22] Biology Department, University of Miami, Coral Gables, FL, USA. ✉email: ajduque@unal.edu.co

Tropical and subtropical ecosystems are believed to account for nearly 70% of all the carbon (C) sequestered by Earth's forests[1]. However, estimates of tropical C uptake are largely based on studies of lowland ecosystems[2–4]. Limited climatic variation in lowland tropical forests[4] hampers our ability to extrapolate the observed trends and purported drivers of C dynamics into areas with greater environmental heterogeneity and steeper climatic gradients, such as montane forests. Quantifying the extent to which tropical and subtropical montane forests contribute to C uptake is essential for generating comprehensive estimates of global C cycling, as well as for helping to motivate the preservation of these forests and the multiple ecosystem services[5] that these biodiversity hotspots[6] provide. Furthermore, there is a pressing need to identify the ecological factors that drive large-scale changes in the amount of C stored in the living aboveground biomass (AGB) of tropical montane forests (hereafter AGC), to increase our predictive understanding of how these systems, which are already highly threatened by anthropogenic exploitation, can contribute to future C storage and cycling[4].

The Andes are the world's longest mountain range and its "hottest hotspot" of biodiversity[7]. Yet estimates of Andean C stocks remain sparse[8–10] and poorly quantified at regional scales. Unique among the world's continents, many of the most-populous cities in South America (e.g. La Paz, Cuzco, Quito, Bogotá, Santiago) are located in the mountains at elevations above 500 m asl[11]. This feature reflects both the distributions of pre-Hispanic indigenous populations and patterns of post-Spanish colonization, and has led to a long legacy of anthropogenic disturbances in Andean ecosystems. Ongoing human activities, together with the inherent instability of steep mountainous terrains, have created a heterogeneous mosaic of natural forests with different levels of disturbance throughout Andes[12]. Additionally, global warming is generating directional changes in forest composition via the upslope shift of some species' ranges[13]. These rapid compositional changes raise the prospect of considerable C losses at the lagging edge of species' ranges through the elevated mortality of large adult trees in areas that become too hot and/or dry[10,14]. The resultant biomass loss may be only partially and slowly offset through the increased recruitment and growth of other more-thermophilic species migrating upslope. This process of compositional changes due to upslope species migrations, known as thermophilization[13], is expected to directly enhance forest disturbance and indirectly increase overall mortality, and thus may drive net losses of AGC. Disentangling the relative importance of disturbance and climate change as concurrent, and potentially interacting, drivers of AGC dynamics in Andean forests is essential for understanding and predicting the role of tropical forests in global C cycling. Although both human-driven disturbances and climate change are likely to continue through the foreseeable future[15], $CO_2$ fertilization and warming could positively influence C uptake upslope in montane ecosystems.

The AGC stocks and productivity of Mature Andean forests are typically characterized as decreasing monotonically with elevation, due largely to colder temperatures and harsher climates in the highlands[8]. There is growing recognition, however, that patterns of AGC can be complex and that both C productivity and storage in tropical montane forests can be impacted by multiple biotic factors. In particular, belowground symbiotic root associations (SRA) have been increasingly recognized as key drivers of forest dynamics and soil C stocks[16,17], but to date their role in Andean forests has mostly been evaluated with global models at relatively coarse geographic resolutions[18,19]. Likewise, there is emerging interest in the importance of evolutionary history and phylogenetic diversity (PDz) in determining patterns of AGC stocks and

productivity[20], and specifically the increases in tree size and biomass that occur in many cold, high-elevation tropical montane forests[10,21]. The evolutionary dimension of biodiversity may affect ecosystem functioning via two mechanisms: (i) niche complementarity in resource use by functionally different clades, allowing for increased productivity of species assemblages[22]; and (ii) selection of functionally-redundant clades promoting the dominance of the most-productive species[23]. Selection effects are predicted when niche conservatism results in the maintenance of evolutionary traits that enhance productivity and resource acquisition[24]. Directly quantifying and disentangling the relative contributions of biotic and abiotic factors on AGC dynamics in Andean forests will significantly enhance our ability to forecast the future composition and function of these forests.

In this study, we analyze AGC dynamics by synthesizing data from an extensive network (Red de Bosques Andinos; RBA)[25] of 119 forest-monitoring plots across five countries spanning the subtropical and tropical Andes (Fig. 1). The AGC dynamics of each plot are characterized using the annualized values (Mg C ha$^{-1}$ y$^{-1}$) of AGC mortality, AGC productivity, and AGC net change[3] (see Methods). We use both structural equation modeling (SEM)[26] and an Information Theory (IT) natural model-averaging technique[27] to identify the dominant climatic and biotic drivers of AGC dynamics (Supplementary Table 1) across the Andean forest plot network. The regional climate is characterized through Principal Component Analyses of temperature (PCA$_{temp}$) and precipitation (PCA$_{prec}$) applied to suites of climate variables[28]. The biotic explanatory variables are: (i) the Thermophilization Rate (TR)[13,14]; (ii) symbiotic root associations (SRA—characterized by the log-transformed ratio of arbuscular mycorrhizal (AM) to ectomycorrhizal (EcM) trees)[19]; (iii) plant phylogenetic diversity (PDz)[20]; and (iv) a size-dependent mortality parameter (β) derived from the probability of tree death as a function of DBH was employed as a measure of post-disturbance recovery[29]. Finally, we estimate the extent of forest cover and the rate of forest loss for the Andes mountains between the years 2003 and 2014[30,31] within elevation bands that represent recognized habitat zones[25] (see Methods). We show a comprehensive continental-scale assessment of patterns and drivers of AGC dynamics in mature Andean forests. Our objectives in this assessment are to (1) determine whether Andean forests currently function as AGC sinks or sources; (2) identify the main ecological drivers of AGC gains and losses in Andean forests; and (3) estimate changes in total AGC stocks across subtropical and tropical Andean forests over the past two decades. In the subtropical and tropical Andean forests, we find differences in the net rates of AGC productivity and mortality, which are driven by a combination of abiotic and biotic factors. Overall, Andean forests are strong sinks for carbon in spite of the fact that global warming appears to be controlling increased mortality of large trees in hotter portions of their species' ranges. Reducing deforestation and increasing restoration will increase AGC stocks in the Andes by expanding forest area and allowing upward species migrations to support biomass gains.

## Results

**Drivers of aboveground carbon dynamics in Andean forests.** The mean net change in aboveground biomass (AGB) across our network of 119 subtropical and tropical Andean forest plots was 1.44 ± 0.18 Mg ha$^{-1}$ y$^{-1}$ (mean ± SE). This represents a net increase in AGC of 0.67 ± 0.08 Mg C ha$^{-1}$ y$^{-1}$, corresponding to a proportional increase of 1.01 ± 0.13% C ha$^{-1}$ y$^{-1}$ (Fig. 2a, b, Supplementary Table 2). This net increase was the result of greater AGC productivity (Fig. 2c, d) than AGC mortality (Fig. 2e, f). According to both the SEM and IT analyses, when analyzing just abiotic variables (PCA$_{temp}$1, PCA$_{temp}$2, AGC1),

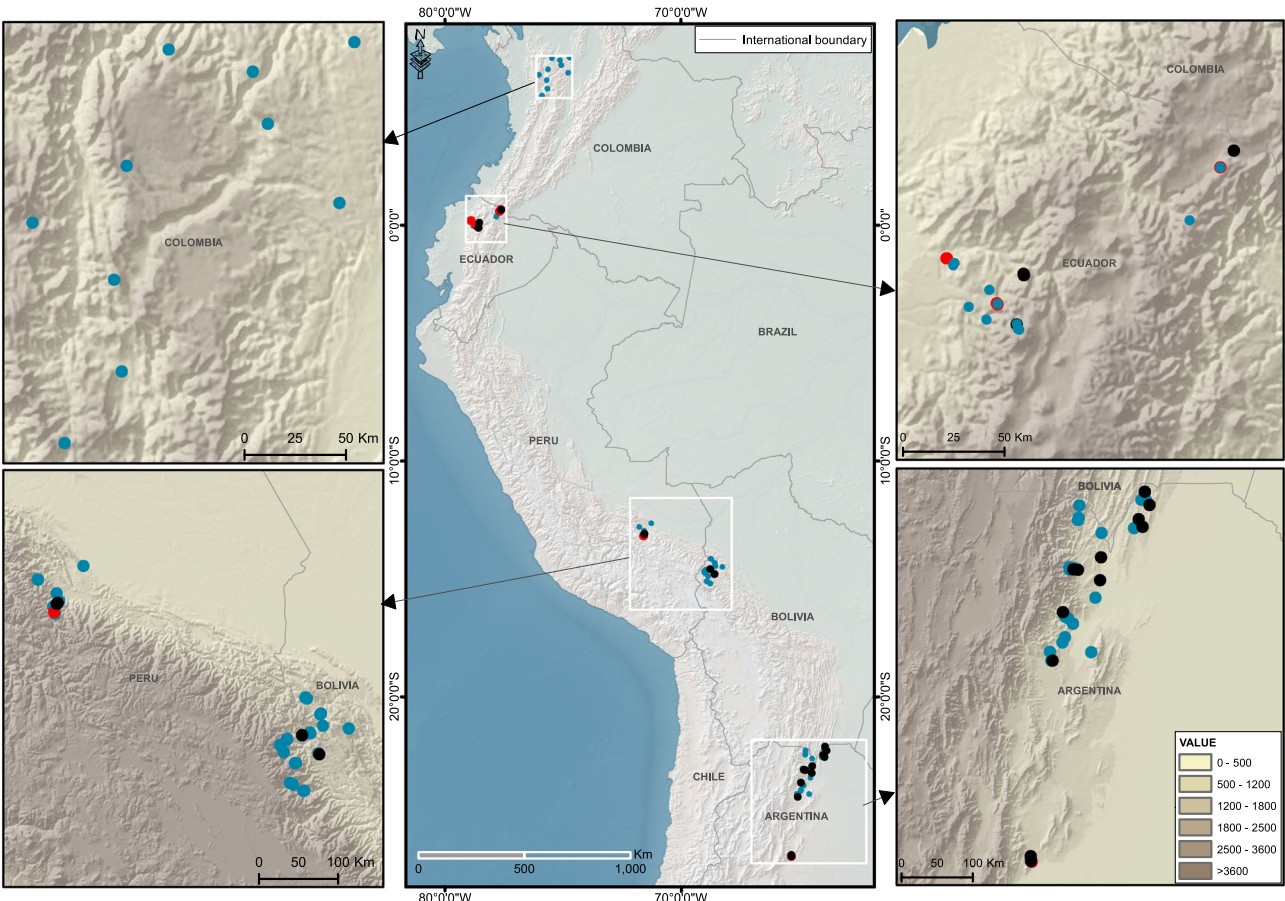

**Fig. 1 Forest plots distribution in the Andean region.** Distribution of the 119 forest-monitoring plots along latitudinal and elevational gradients in the subtropical and tropical Andes. The black points represent plots in which the aboveground carbon net change (AGC net change) was negative ($<0$ Mg ha$^{-1}$ y$^{-1}$). Blue points: $0 \leq$ AGC net change $\leq 3$ Mg ha$^{-1}$ y$^{-1}$. Red points: AGC net change $>3$ Mg ha$^{-1}$ y$^{-1}$.

only PCA$_{temp}$2 significantly explained the AGC net change (R$^2$$_{SEM}$ = 0.18; see Methods). When both biotic and abiotic variables were considered, the size-dependent mortality parameter ($\beta$) was included in the best-fitting SEM and IT models while AGC1 was significant only in the SEM (R$^2$$_{SEM}$ = 0.32) (Fig. 3a; Supplementary Table 3). Surprisingly, none of the analyses indicated a significant association between AGC net change and plot elevation (Fig. 2).

Climate characterization (Supplementary Fig. 1) showed a high correlation between temperature and precipitation (Supplementary Fig. 2). Therefore, we only used the PCA$_{temp}$1 and PCA$_{temp}$2 axes as explanatory variables in the subsequent models to represent an overall climate gradient of elevation (r = −0.97, $p < 0.001$) and latitude (r = 0.78, $p < 0.001$), respectively (Supplementary Fig. 3). The PCA$_{temp}$2 was largely defined by a decline in annual temperature range, an increase of minimum temperature of the coldest month, and an increase of total annual precipitation (Supplementary Table 4). The PCA$_{temp}$2, characterized the increase of net change in AGC along the south-north climate gradient (Fig. 2). The correlation between PCA$_{temp}$2 and PCA$_{prec}$1,2 (Supplementary Fig. 2) emphasizes the positive relationship between AGC net change and mean annual temperature, and a negative relationship between AGC net change and rainfall seasonality. Of the biotic variables analyzed, AGC net change was negatively associated with $\beta$ (Fig. 3a, Supplementary Table 3), which in turn was primarily shaped by TR (Fig. 3a). The strong effect of $\beta$ on the AGC net change suggests that much of the recorded increase in AGC net change in tropical and subtropical Andean forests over the past two decades

has been due to post-disturbance growth of larger trees in plots where post-disturbance recovery enhanced the probability of mortality for small trees (Supplementary Fig. 4). This positive effect of $\beta$ on AGC net change (Supplementary Fig. 5a) was lower ($0.48 \pm 0.09$ Mg ha$^{-1}$ y$^{-1}$) but still evident when the 30 plots below the 0.25 quartile of $\beta$ (i.e. the plots with the strongest signal of post-disturbance recovery) were excluded.

Analyses of just the abiotic variables indicate that AGC productivity was significantly associated with AGC1 and PCA$_{temp}$2 in both the IT and SEM analyses (R$^2$$_{SEM}$ = 0.46). When the biotic variables were included, AGC productivity was negatively correlated with PDz, but positively correlated with SRA (R$^2$$_{SEM}$ = 0.50) (Fig. 3b; Supplementary Table 3). Collectively, this indicates that Andean forest AGC productivity is higher near the equator in plots with higher initial AGC stocks, greater proportional abundance of AM fungal associations, and lower phylogenetic diversity (Fig. 3b; Supplementary Table 3).

Unlike AGC net change and AGC productivity, AGC mortality was not correlated with climate when assessed across all plots (Fig. 2e, f). Indeed, when considering just the abiotic variables, AGC1 was the only variable that significantly explained AGC mortality in both the IT and SEM analyses (R$^2$$_{SEM}$ = 0.19). When all the biotic variables were also considered, $\beta$ emerged as another significant explanatory variable in both analyses (R$^2$$_{SEM}$ = 0.47) (Fig. 3c; Supplementary Table 7). Under the IT approach, TR and PDz were also selected as significant explanatory variables, despite the positive correlation between $\beta$ and TR. In contrast, the SEM analysis found SRA, but not TR ($p = 0.06$) or PDz (Fig. 3c), to be a direct significant driver of AGC mortality (TR was

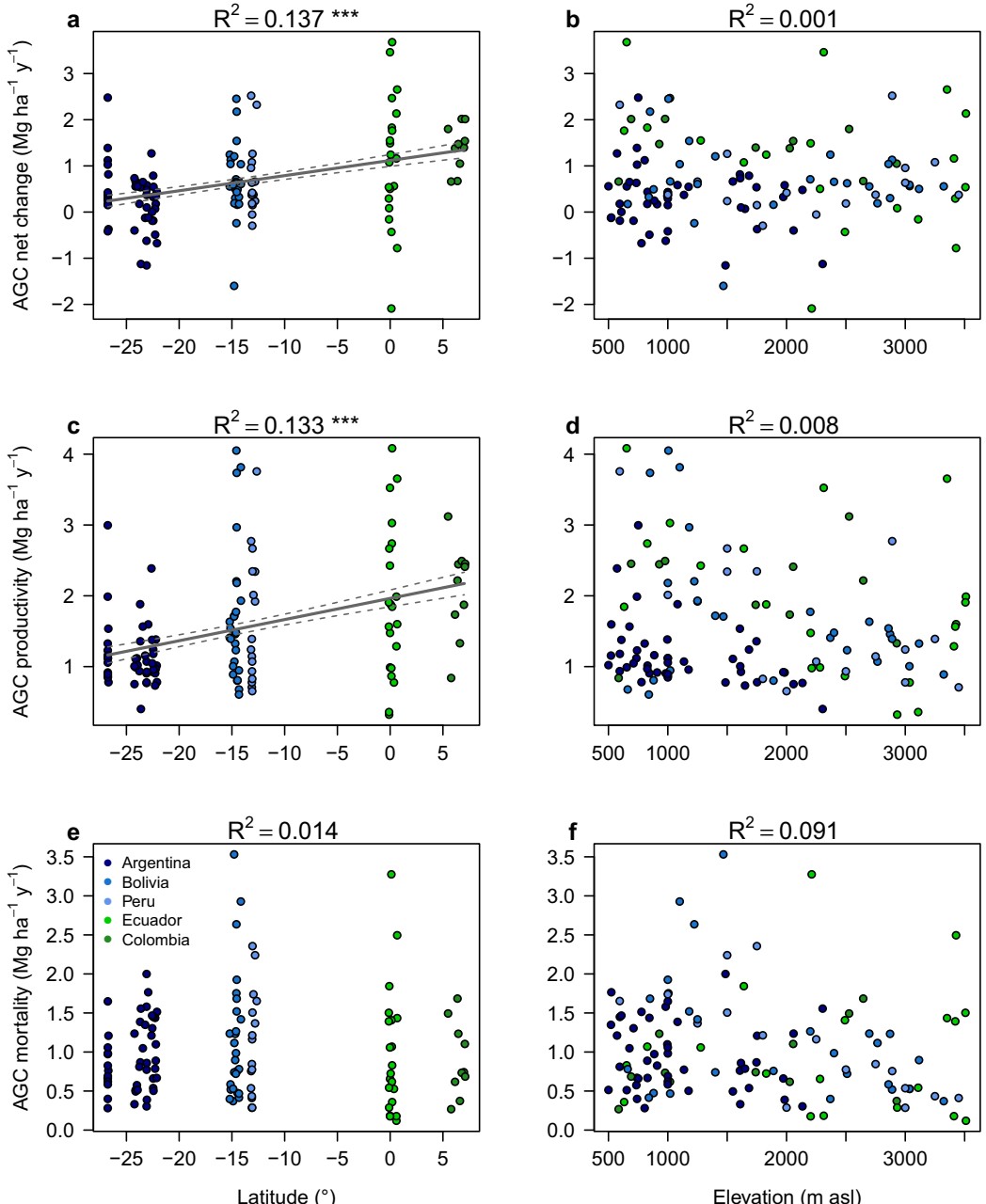

**Fig. 2 Aboveground carbon dynamics along elevational and latitudinal gradients.** Generalized additive models (GAMs) assessing the changes of the aboveground carbon (AGC) dynamics in relation to latitude (°) (**a**, **c**, **e**) and elevation (m asl) (**b**, **d**, **f**), across 119 forest-monitoring plots in the subtropical and tropical Andes. Negative latitudinal values represent plots in the Southern hemisphere and positive latitudinal values represent plots in the Northern hemisphere. Solid lines represent the models that were statistically significant and da shed lines represent 95% confidence intervals. *$P \leq 0.05$; **$P \leq 0.01$; ***$P \leq 0.001$. AGC net change = aboveground carbon net change (Mg C ha$^{-1}$ y$^{-1}$). AGC productivity = aboveground carbon productivity (Mg C ha$^{-1}$ y$^{-1}$). AGC mortality = aboveground carbon mortality (Mg C ha$^{-1}$ y$^{-1}$). $R^2$ = Coefficient of determination of the model.

indirectly associated with AGC mortality due its significant correlation with β). Overall, based on the two analytical methods applied (SEM and IT), the strongest and most consistent driver of AGC mortality was β, a result that may reflect the fact that both AGC mortality and β are size-dependent and thus inherently correlated (i.e. large trees contain more C than smaller trees and thus their disproportionate loss leads to greater declines in total AGC). When β was excluded, TR became one of the most important explanatory factors explaining AGC mortality in the IT models (Supplementary Table 5; Supplementary Fig. 6). Hereafter, we discuss the influence of AGC1, β, and TR as the main drivers of AGC mortality.

**Carbon balance and forest cover change.** The estimated mean AGC stock across the Andean forests for the year 2003 represents a total C stock of 3.83 Pg (3.31–4.34 95% CI) that increased up to 4.12 Pg C (3.43–4.81 95% CI) in 2014. AGC stocks at both the plot (Supplementary Fig. 7) and landscape (Table 1) levels decreased significantly with elevation, with nearly half of the total Andean C stocks (1.71 Pg C) located in the foothills (500–1200 m asl) (Table 1). The total AGC forest cover weighted mean rate change of 0.66 (0.34–0.96 95% CI) Mg C ha$^{-1}$ y$^{-1}$ found between 2003 and 2014, was almost the same than the observed 0.67 ± 0.08 Mg C ha$^{-1}$ y$^{-1}$ AGC net change at the plot level in nearly two decades. In Andean forests, this rate of change implies an

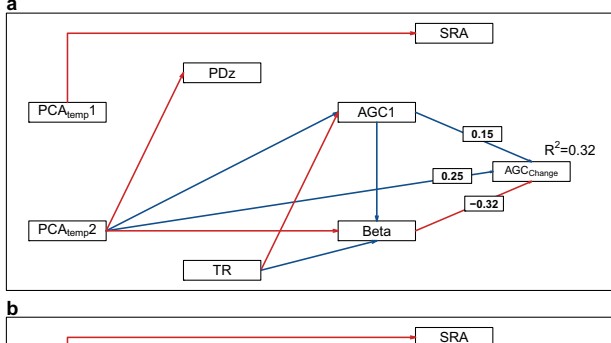

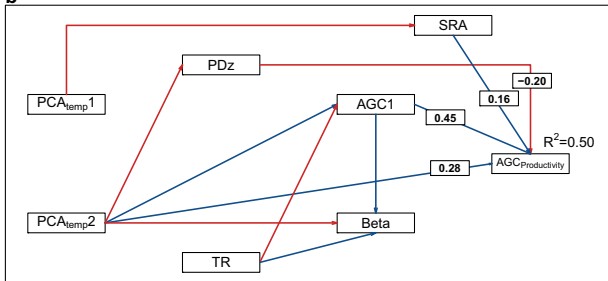

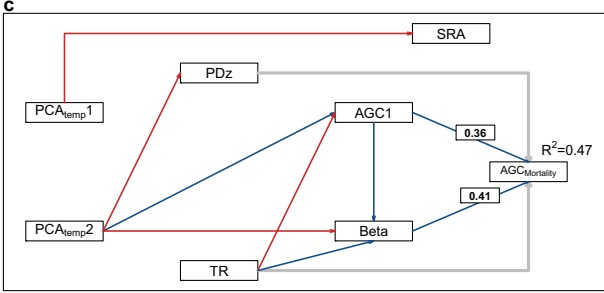

**Fig. 3 Drivers of aboveground carbon dynamics in the Andes.** Structural equation models (SEMs) used to evaluate the effects of climate ($PCA_{temp}1$ and $PCA_{temp}2$), initial aboveground carbon stock in each plot (AGC1; Mg C ha$^{-1}$), thermophilization rate (TR; °C y$^{-1}$), symbiotic root associations (SRA = ln(AM/EcM)), the standardized effect size of the phylogenetic diversity (PDz), and the size-dependent probability of mortality (β) on aboveground carbon dynamics. $AGC_{change}$ = aboveground carbon net change (Mg C ha$^{-1}$ y$^{-1}$) (**a**). $AGC_{productivity}$ = aboveground carbon productivity (Mg C ha$^{-1}$ y$^{-1}$) (**b**). $AGC_{mortality}$ = aboveground carbon mortality (Mg C ha$^{-1}$ y$^{-1}$) (**c**). Red arrows indicate negative relationships and black arrows indicate positive relationships. Grey arrows represent significant ($P \leq 0.05$) relationships in the IT models (Supplementary Table 7), but not in the SEM models. The values over the arrows are the associated linear coefficients of the explanatory variables found to be significant. R$^2$ = Coefficient of determination of the overall model.

estimated total AGC sink of 0.027 Pg C y$^{-1}$ (0.011–0.042 95% CI) (Table 1). This increase of C storage is in spite of a 4.2% reduction in total forest cover (12,687 km$^2$), which is estimated to have released 0.33 Pg C0$_2$ equivalent (0.28–0.37 95% CI) between 2003 and 2014.

## Discussion

Our results indicate that Andean forests are acting as globally significant AGC sinks. Our estimate of net annual change in aboveground carbon (AGC net change) for tropical and sub-tropical Andean montane forests (0.67 ± 0.08 Mg C ha$^{-1}$ y$^{-1}$) compares with much more uncertain change rates previously estimated using different and substantially smaller datasets of secondary (0.23 ± 0.87 Mg C ha$^{-1}$ y$^{-1}$; mean ± SD) and old growth (0.82 ± 0.37 Mg C ha$^{-1}$ y$^{-1}$) montane forests in North and South America[2], which are the default values employed by the intergovernmental panel on climate change (IPCC). Notably, the strength of the C sink in Andean forests is greater than the 0.42 Mg ha$^{-1}$ y$^{-1}$ of C sink estimated for lowland Amazonian forests over a similar time period[3,4] despite the lower AGC stocks of montane forests. Taken together, our results indicate that the Andes are similar to other tropical forests in that they are acting as AGC sinks, but the overall relative strength of the Andean C sink (1.01% annually) is even stronger than that of mature lowland tropical forests in Amazonia[3], Africa[4], or southeast Asia[32]. The continued net uptake of C in Andean forests will become even more important as the C sinks in lowland tropical forests become increasingly saturated[4].

The relationships between AGC mortality and AGC net change with the size-dependent mortality patterns indicate a significant influence of disturbance and self-thinning[33] (Supplementary Fig. 4) on the carbon dynamics of Andean forests (Supplementary Fig. 5a). When we focused on just the plots with no or little influence of post-disturbance competitive thinning (i.e. β between the 0.25 and 1.0 quartiles), the AGC net change (0.48 ± 0.09 Mg ha$^{-1}$ y$^{-1}$) remained positive and was more similar to that of "intact" lowland forests. In other words, around 30% of the net AGC uptake in Andean forests may be attributable to recovery from past disturbance, while the remaining appears to be due to other factors, such as CO$_2$ fertilization and temperature increase[34]. A significant change in stem density accompanied by no increase in average tree wood density with AGC net change (Supplementary Fig. 8) mirrors the mature lowland forests where a similar trend has previously been interpreted as a signal of CO$_2$ fertilization causing increased AGC (see Fig. 3 in ref. [3]). However, we readily acknowledge that the quantification of tree mortality and AGC dynamics is highly dependent on sampling intensity[35], which emphasizes the need for future studies to use larger sample sizes to help differentiate the

**Table 1 Andean forest cover and total Above Ground Carbon (AGC) stocks (Pg) per elevation band range, estimated for the years 2003 and 2014.**

| Elevation range (m asl) | AGC stock initial census (Mg C ha$^{-1}$) | AGC stock final census (Mg C ha$^{-1}$) | Forest cover (km$^2$) | | Total AGC stocks (Pg) | |
|---|---|---|---|---|---|---|
| | | | 2003 | 2014 | 2003 | 2014 |
| 500–1200 | 71.89 (62.97–81.37) | 88.85 (70.79–105.86) | 237,820.21 | 230,115.49 | 1.71 (1.50–1.94) | 2.04 (1.63–2.44) |
| 1200–2000 | 73.85 (65.11–81.83) | 73.25 (65.47–80.93) | 175,122.11 | 171,684.02 | 1.29 (1.14–1.43) | 1.26 (1.12–1.39) |
| 2000–2800 | 63.33 (49.72–76.63) | 62.96 (52.16–74.09) | 94,961.05 | 93,844.90 | 0.60 (0.47–0.73) | 0.59 (0.49–0.70) |
| 2800–3600 | 56.69 (50.52–63.12) | 59.60 (49.13–74.09) | 39,172.57 | 38,744.63 | 0.22 (0.20–0.25) | 0.23 (0.19–0.29) |
| Weighted mean | 69.94 (60.46–79.39) | 77.17 (64.24–89.97) | 547,075.95 | 534,389.05 | 3.83 (3.31–4.34) | 4.12 (3.43–4.81) |

AGC stocks are estimated at initial and final censuses multiplied by forest area for 2003 and 2014, respectively (mean and bootstrapped 95% CI) (see Main text). The total C forest stock is the total estimated amount for each elevational band and the whole subtropical and tropical Andean region. The weighted mean represents the overall mean weighted by forest cover according to either the initial (2003) or final (2014) census.

drivers of biomass dynamics in complex, tropical mountain forest ecosystems.

Remarkably, the rapid AGC net gains in Andean forests occurred in spite of changes in tree species composition caused by warming (i.e. the positive relationship between β and TR). Indeed, the high AGC mortality found in forests between 1000 and 1800 m asl (Fig. 2f), where previous studies have shown large variation in species' thermal tolerances[36], seems to be largely due to the influence of systematic changes in species composition (Supplementary Table 5). The indirect negative effect of TR on C stocks likely reflects the loss of individuals in the hotter portion of their species' thermal ranges[13,14], many of which represent large-statured early/intermediate-successional species with relatively low wood density (Supplementary Fig. 5b, c). It is possible that these species are particularly sensitive to either carbon starvation or hydraulic failure due to the increases in temperature and vapor pressure deficit associated with global warming[15]. It is likely, however, that declines in AGC net change due to thermophilization is only a transient phenomenon, since the dead trees should eventually be replaced by large and more-productive species with hotter thermal optima that can take advantage of increasing temperatures, $CO_2$, and N availability. Indeed, the future capacity of Andean forests to store C in the living aboveground biomass will depend substantially on the ability of lowland thermophilic species to move into higher elevations and replace those species that are lost to heat stress. Such species migrations depend critically on ecosystem connectivity to permit the diaspora of tree species (through seed dispersal) and their above- and belowground symbionts.

In marked contrast to recent findings from lowland Amazonia[20], AGC productivity and phylogenetic diversity (PDz) are negatively correlated in Andean forests. Our analyses revealed a strong influence of large-statured, mostly tropical-affiliated, clades (e.g. Fabaceae, Lauraceae, and Moraceae) as the determinants of 46% of the total AGC productivity in both subtropical and tropical Andean forests at lower elevations (<2000 m asl), while others clades (e.g. Cunoniaceae, Melastomataceae, and Clusiaceae) dominated AGC productivity (51.5%) at higher elevations (>2800 m asl). At intermediate elevations (1000–2800 m asl), Lauraceae was the dominant clade in tropical sites, while Myrtaceae and Podocarpaceae were the dominant clades in the subtropical sites (Supplementary Table 6). These contrasting patterns suggest that selection effects[23] and the conservation of large stature within just a few key clades with different evolutionary histories[21], play an important role in driving AGC productivity in Andean forests (Supplementary Fig. 9). Understanding the traits that confer competitive advantages to these clades at different elevations will inform both conservation and restoration of AGC stocks in Andean forests.

The effect of symbiotic root association (SRA) on AGC productivity in Andean forests is consistent with growing evidence that mycorrhizal associations act as important ecological drivers of forest C dynamics at large spatial scales[16,19]. The positive association between AGC productivity and SRA parallels temperate systems where faster nutrient cycling in AM-dominated forests drives more rapid plant growth[16]. AM fungal symbionts may be particularly important for enhancing phosphorus (P) and nitrogen (N) uptake in Andean forests, especially at low temperatures (Supplementary Fig. 10) where low nutrient availability can significantly limit plant growth[16,37]. Since climate is a key determinant of mycorrhizal type at both continental[16] and global scales[18], it is likely that changing environmental conditions in the Andes will also alter soil-nutrient cycling rates and C storage. In particular, under enriched $CO_2$ levels, the productivity of AM-dominated forests is significantly lower than that of EcM-dominated forests due to greater N limitation[38], suggesting that

the positive association we observed between AGC productivity and AM tree abundances may weaken in the future.

The increasing AGC stocks in remaining forests more than offset the estimated C emissions due to deforestation and forest loss, resulting in a net total uptake of 0.027 Pg C y$^{-1}$ in the Andes (Table 1). Indeed, the total AGC stock and carbon uptake of Andean forests would be even greater if forest regrowth (estimated as ~500,000 ha between 2001 and 2014, mostly in abandoned pastures and agricultural lands at mid-elevations[11]) were included in our estimates. The strong capacity of montane forests to gain AGC[39], along with the expected long-term gains due to upslope species' migrations, mean that post-disturbance forest recovery can contribute substantially to greater C storage in the Andes. Together, protecting natural forests and increasing restoration efforts[40] can help to secure the Andes' contribution as a critical global refuge for both C and biodiversity.

Overall, Andean forests represent globally significant AGC sinks, and have the potential to serve as important future C refuges. Indeed, due to the declining strength of carbon sinks in lowland tropical forests[4], the importance of montane systems for carbon management is increasing. It is therefore critical to stop and reverse the loss of Andean forests, particularly within the 500–1800 m asl elevation band which accounts for >60% of recent deforestation[11,30]. As well as impacting forest area and carbon storage directly, deforestation at mid-elevations can disrupt the functional connections between lowland and highland forests. Safeguarding Andean forest connectivity will be critical not only for biodiversity conservation per se but also for protecting and enhancing future carbon storage.

## Methods

**Study area.** This study was conducted using tree census data collected from 119 forest inventory plots (73 tropical, 46 subtropical) situated across a latitudinal range of 7.1°N (Colombia) to 27.8°S (Argentina), a longitudinal range of 79.5° to −63.8° W, and an elevation range of 500–3511 m asl (Fig. 1). The mean annual temperature (MAT) of plots ranged from 7.3 to 23.8 °C (mean = 16.7 ± 4.1 °C; mean ± SD) and mean annual precipitation (MAP) of the plots ranged from 608 to 4313 mm y$^{-1}$ (mean = 1405.0 ± 623.9 mm y$^{-1}$) (External Databases 1). The number of plots sampled in each country was: Argentina = 46, Bolivia = 26, Peru = 16, Ecuador = 21, and Colombia = 10 (Fig. 1). The 119 forest plots ranged in size from 0.32 to 1.28 ha and represent a cumulative sample area of 104.4 ha (horizontal areas corrected for slope) that contain more than 63,000 trees with a diameter at breast height (DBH, 1.3 m) ≥10 cm (External Database 1). Ninety-four of the plots (79.0%) were ≥1 ha in size. Neither secondary forests nor plantations were included. However, only seven of the plots (five in Argentina and two in Bolivia) were located in forests >100 km² in extent[41], which suggests that at least the edges and borders of some plots could have experienced some degree of disturbance or degradation. All plots were censused at least twice between 1991 and 2017 (census intervals ranged between 2 and 9 years).

In each plot, we tagged, mapped, measured, and collected vouchers of all trees and palms (DBH ≥ 10 cm). DBH was measured 50 cm above buttresses or aerial roots when present (where the stem was cylindrical). During the second or subsequent set of censuses, DBH growth, recruitment, and mortality were recorded. In cases where the recorded DBH growth of the second census was less than −0.1 cm y$^{-1}$ or greater than 7.5 cm y$^{-1}$, the DBH of the second census was augmented/reduced in order to match these minimum/maximum values[42]. To homogenize and validate species names of palms and trees recorded in each country and plot, we submitted the combined list from all plots to the Taxonomic Name Resolution Service (TNRS; http://tnrs.iplantcollaborative.org/) version 3.0. Any species with an unassigned TNRS accepted name or with a taxonomic status of 'no opinion', 'illegitimate', or 'invalid' was manually reviewed. Families and genera were changed in accordance with the new species names. If a full species name was not provided or could not be found, the genus and/or family name from the original file was retained.

**Aboveground carbon stocks.** The aboveground biomass (AGB) of each tree was estimated using the allometric equation proposed by Chave et al[43]., defined as: $AGB = 0.0673 \times (WD \times DBH^2 \times H)^{0.976}$ where AGB (kg) is the estimated aboveground biomass, DBH (cm) is the diameter of the tree at breast height, H (m) is the estimated total height, and WD (g cm$^{-3}$) is the stem wood density. To estimate WD, we assigned the WD values available in the literature[44] to each species found in each plot. In cases where we could not assign a WD value at the species level, we used the average value at the genus- or family level. For unidentified individuals, we

used the average WD value of all other species in the plot. Tree height (H) was estimated (see below) based on the heights measured on a subset of the individual stems in each plot using digital hypsometers or clinometers. The estimated AGB of each tree was then converted to units of aboveground carbon (AGC) by applying a conversion factor of 1 kg AGB = 0.456 kg C[45]. The AGC per ha was then determined by converting kg to Mg, summing the values for all trees in a plot, and extrapolating or interpolating to a sample area of 1 ha.

Estimates of AGB and AGC are highly dependent on tree height. Unfortunately, tree height was difficult or impossible to measure on all stems due to physical and logistical constraints. Therefore, we estimated the height of each stem based on allometric relationships between DBH and tree height that we developed for each plot based on height and DBH measurements taken on a subset of individuals. Although the AGB/AGC estimates are only for trees with DBH ≥ 10, we used trees with DBH ≥ 5 cm to construct the H:DBH models when possible in order to be as comparable as possible with the existing pantropical H:DBH models[46]. In total, 44,442 trees had their heights measured in the field and were employed to construct the H:DBH models. The percentage of trees with direct field measurements of H (DBH ≥ 5 cm) in each country was: Argentina = 19%, Bolivia = 98%, Peru = 96%, Ecuador = 97%, and Colombia = 46%. In Argentina, 32 of 46 plots did not have any field measurements of H, while all plots in all other countries had field measurements of H for at least a subset of trees.

We tested and compared the expected effects of using H:DBH models constructed using the local (plot), country, or pantropical (regional) level data. To select the best model to estimate H from DBH at the plot and country level, we used the function modelHD available in the BIOMASS package for R[47]. We chose the best allometric model from four candidate models (two log-log polynomial models, the three-parameter Weibull model, and a two-parameter Michaelis-Menten model (Supplementary Table 7)) by selecting the model with the lowest RSE and bias (Supplementary Table 8). At the regional level, we used a pantropical model[46]. The use of country and pantropical H:DBH allometries underestimates tree heights in the lowlands and overestimates tree heights in highlands, thereby homogenizing AGB estimates along elevational gradients[10,48] (Supplementary Figs. 11, 12, 13). Using plot level allometries eliminates this problem. However, in the 32 plots in Argentina where we had no information about tree height, we used the country-level H:DBH model developed with the data available in the remaining 14 plots to estimate the height of each tree, which could have homogenized the AGC estimates along the Argentinian elevational gradient (Supplementary Figs. 11, 12, 13).

**Aboveground carbon dynamics**. The AGC dynamics of each plot was estimated from the annualized values of AGC mortality, AGC productivity (AGC change due to recruitment + growth), and AGC net change[3]. The calculations of the separate AGC dynamic components was performed as follows: (i) AGC mortality (Mg ha$^{-1}$ y$^{-1}$) = the sum of the AGC of all individuals that died between censuses divided by the time between measurements. (ii) AGC recruitment (Mg C ha$^{-1}$ y$^{-1}$) = the sum of the AGC of individuals that recruited into DBH ≥ 10 cm between censuses divided by the time between measurements. However, for each tree recruited (DBH ≥ 10 cm), we subtracted the corresponding AGC associated with a tree of 9.99 cm (i.e. just below the detection limit) in order to avoid overestimations of the overall increase in AGC due to recruitment[49]. (iii) AGC growth (Mg ha$^{-1}$ y$^{-1}$) = the sum of the increase in AGC of all individuals with DBH ≥ 10 cm that survived between censuses divided by the time between censuses. (iv) AGC net change (Mg ha$^{-1}$ y$^{-1}$) = the difference between AGC stock in the last census (AGC$_{final}$) and AGC stock in the first census (AGC1) divided by the elapsed time (t; in years) between measurements [(AGC net change = AGC$_{final}$ − AGC1)/t]. We recognize that these methods exclude C stored in soils or in belowground tissues[9,48]; however, quantifying just aboveground C stocks and fluxes provides valuable information about the overall status of these forests as net C sinks or sources.

**Climate**. Climate variables at each plot location were extracted from the CHELSA[28] bioclimatic rasters at a resolution of 30-arcsec (~1 km$^2$ at the equator). The climate variables extracted were: Mean Annual Temperature (MAT), Mean Diurnal Range (MDR), Isothermality (Isoth), Temperature Seasonality (TS), Maximum Temperature of Warmest Month (MaxTWarmM), Minimum Temperature of Coldest Month (MinTCM), Temperature Annual Range (TAR), Mean Temperature of Wettest Quarter (MeanTWetQ), Mean Temperature of Driest Quarter (MeanTDQ), Mean Temperature of Warmest Quarter (MeanTWetQ), Mean Temperature of Coldest Quarter (MeanTCQ), Mean Annual Precipitation (MAP), Precipitation of Wettest Month (PWetM), Precipitation of Driest Month (PDM), Precipitation Seasonality (PS), Precipitation of Wettest Quarter (PWetQ), Precipitation of Driest Quarter (PDQ), Precipitation of Warmest Quarter (PWarmQ), Precipitation of Coldest Quarter (PCQ). We separated all variables associated with temperature (°C) from those associated with precipitation (mm y$^{-1}$) and applied a Principal Component Analysis (PCA) to the 11 variables associated with temperature (PCA$_{temp}$) and a separate PCA to the eight variables associated with precipitation (PCA$_{prec}$). The first two principal components of both PCA$_{temp}$ and PCA$_{prec}$ (four PCA axes in total) were selected for use in subsequent analyses. Plot elevations were estimated based on their coordinates and the SRTM 1 ArcSec Global V3 (https://lta.cr.usgs.gov) 30 m resolution digital elevation model (DEM).

PCA$_{temp}$1 (Supplementary Fig. 1a) explained 53.0% of the total variance of the temperature variables and had high loading from Isothermality and Maximum Temperature of Warmest Month, which was primarily associated with changes in elevation (r = −0.97, p < 0.001) (Supplementary Fig. 3b). PCA$_{temp}$2, explained 45.2% of the total variance of the temperature and had high loading of Temperature Annual Range and Minimum Temperature of Coldest Month. The PCA$_{temp}$2 was primarily associated with changes in latitude (r = 0.78, p < 0.001) (Supplementary Fig. 3c). PCA$_{prec}$1 (Supplementary Fig. 1b) explained 68.9% of the total variance of the precipitation variables was highly loaded by Mean Annual Precipitation and Precipitation Seasonality, and was primarily associated with changes in latitude (r = 0.68, p < 0.001) (Supplementary Fig. 3e). PCA$_{prec}$2 (Supplementary Fig. 1b) explained 26.3% of the total variance of the precipitation variables was highly loaded by Precipitation Seasonality and Precipitation of Driest Month, and was also primarily associated with changes in latitude (r = 0.64, p < 0.001) (Supplementary Fig. 3g).

PCA$_{prec}$1 was negatively correlated with PCA$_{temp}$1 (−0.55, p < 0.001), indicative of the fact that precipitation generally decreased from the lowlands to highlands. Likewise, PCA$_{prec}$2 was negatively correlated with both PCA$_{temp}$1 (−0.46, p < 0.001) and positively with PCA$_{temp}$2 (0.58, p < 0.001). These associations suggest that precipitation seasonality decreases towards the equator, but increases with elevation (Supplementary Fig. 2). Therefore, we only used the PCA$_{temp}$1 and PCA$_{temp}$2 axes as explanatory variables in the subsequent models to represent an overall climate gradient of elevation and latitude, respectively (Supplementary Fig. 3). The use of PCA axes instead of raw climate variables minimizes multicollinearity among variables and better represents the multidimensional gradient of climate that occurs across our study plots.

**Thermophilization rate**. We used the Thermophilization Rate (TR; °C y$^{-1}$)[13,14] as a metric of the direction and rate of compositional changes in the tree communities in relation to species' thermal optima through time. A positive TR indicates an increase in the relative abundances of lowland, thermophilic species (as expected due to the upward shift of species ranges under increasing temperatures) and a negative TR indicates an increase in relative abundances of less-thermophilic, highland species.

To calculate the TR of each plot, we downloaded all available herbarium collection records from the forested regions of tropical Andean countries through the Global Biodiversity Information Facility (GBIF) data portal (www.gbif.org; accessed September 2019). We estimated the mean annual temperatures (MAT) at the collection locations of all specimens by extracting the Bio1 values from the CHELSA climate rasters[28]. For each of the tree species found in the plots and that were represented by 10 or more GBIF records, we estimated their thermal optimum as the mean temperature across all available collection locations within the forested Andean region[50]. For species with <10 available records, we estimated the thermal optimum as the mean temperature across the collection locations of all congeneric individuals within the forested Andean region. Then, for each plot census, we calculated the Community Temperature Index (CTI; °C) as the mean thermal optima of all constituent species weighted by their relative basal areas. Finally, we calculated the annual TR (°C y$^{-1}$) of each plot as the annualized change in the CTI over the entire census period of each plot[13,14].

**Taxonomic and phylogenetic diversity**. All species and genus names were checked and standardized using the Taxonomic Name Resolution Service[51]. In the dataset, 91.3% of stems were identified to species level, 7.3% to genus, 0.8% to family, and 0.6% remained unidentified. To estimate species diversity while accounting for differences in plot size and stem numbers, we used the rarefied Species Richness (SR)[52] at the minimum stem number found per plot (86 individuals). To calculate the phylogenetic diversity *sensu stricto* (PD)[53], we first generated a phylogenetic tree of hypothesized relationships using *phylomatic* for the complete species list as recorded across all 119 plots (excluding unidentified taxa). We used the *bladj* algorithm to date the phylogenetic tree by adjusting phylogenetic branch lengths to respective fossil ages[54]. The PD of each plot community was then calculated as the total sum of the phylogenetic branch lengths connecting the co-occurring species in each plot along the minimum spanning path to the root of the tree.

The observed PD was compared to a null distribution to control for the sampling effects and differences in regional diversity. The null model used an independent swap algorithm that maintained the frequency and richness of species in each plot while randomizing community composition[55]. The standardized effect size of the PD (PDz) was then calculated by subtracting the expected mean PD derived from the null distribution of 999 random draws to the observed PD value in each plot, divided by the standard deviation of the null distribution. This metric was estimated using the *Picante* package in R[56]. SR and PDz were negatively correlated (r = −0.59, p < 0.001). This negative correlation is mainly due to the increased mixture of temperate-affiliated and tropical-affiliated species, both at higher southern latitudes and at higher elevations in the tropics[57,58]. However, SR was never a significant explanatory variable in the models due to the greater relative importance of climate variables, and we only used PDz in the models as the surrogate of complementarity effects.

**Symbiotic root associations.** Symbiotic root associations (SRA), particularly those involving mycorrhizal fungi, are increasingly recognized as important factors influencing plant community productivity and dynamics[16,17]. To incorporate the potential contributions of SRA to AGC stocks and dynamics in subtropical and tropical Andean forests, individuals were assigned an SRA status either as arbuscular mycorrhizal (AM) or ectomycorrhizal fungi (EcM) based on genus- or family-level designations[18]. Overall, the number of genera assigned an SRA status was >90%. We chose these two taxonomic levels for three reasons: (1) assigning species-level taxonomy was difficult for some individuals in our plots; (2) using these higher taxonomic levels greatly increases the ability to provide SRA assignments; and (3) SRA is largely conserved at the genus and family level[59]. Here, we restricted matches for our genera and families to only those present in North and South America in the compiled reference list[18]. Any genera or families lacking symbiotic root assignment were manually checked and, when possible, assigned SRA on the basis of primary literature searches.

We next calculated the relative representation of the two SRA types (AM vs EcM) within each plot. In calculating the relative abundances, we weighted SRA types by the proportional stem number. Since the association between tree species is generally specialized to a single mycorrhizal fungal type (either AM or EcM), the relative representation of the two groups is a good indicator of different modes of nutrient acquisition and soil C stocks[60]. Overall, a high AM to EcM ratio is expected in forest soils that are characterized by faster and inorganic-dominated nutrient cycling, whereas low AM-to-EcM ratios are expected in forest soils with slower and organic-dominated nutrient cycling. Additionally, as the AM-to-EcM ratio decreases, there is generally more C per unit N present in soil, particularly in the upper layers[61]. We used the log-transformed ratio of AM to EcM (e.g. ln (AM/EcM)) as an explanatory variable in our statistical analyses.

**Forest cover and forest loss in the Andes.** We estimated forest cover and forest loss for the Andes mountains between 11°N and 27.3°S and between 82°W and 56°W between the years 2003 and 2014 from Hansen et al. v1.6[30] using Google Earth Engine[31]. We excluded pixels with forest cover <70% from the analysis. We further limited our study area by masking out forests with less than 700 mm of annual rainfall to represent tropical mountain evergreen conditions[62] using Worldclim 2.0 dataset[63], and forests inside terrestrial ecoregions[64]. The time frame of forest change assessed was defined by the median of the year of plot censuses carried out between 1991 and 2009 (2003) and the median of the year plot censuses carried out between 2010 and 2019 (2014). The patterns of forest cover and forest loss were then summarized within four elevation bands that represent recognized habitat zones:[25] (i) 500–1200 m, (ii) 1200–2000 m, (iii) 2000–2800 m, and (iv) 2800–3500 m.

To define the initial (2003) mean AGC carbon stock in each elevational band we used the initial AGC stocks (first plot census) before 2009 (including it). To define the final (2014) mean AGC carbon stock in each elevational band we used the final AGC stocks (last plot census) after 2009 (excluding it). We did not use the plot AGC net change rates due to the high asynchrony among census' dates. That said, in some countries the last (final) census in a particular plot could almost always be before 2010 (e.g. Peru), while in others the initial census was always after 2010 (e.g. Ecuador).

We used bootstrapping to assess the mean and 95% confidence intervals (CI) of AGC stocks in each elevation band for both the initial (2003) and final (2014) years defined to assess changes in forest cover. In each elevational band, total AGC stocks were calculated as the product of the mean AGC stock in the initial/final census and the forest cover at 2003/2014, respectively. The overall mean AGC stock for Andean forests along the whole elevational gradient was quantified as the forest cover weighted mean of AGC stocks across elevational bands. Finally, the net AGC (Mg C y$^{-1}$) balance was calculated as the difference between the total AGC stocks in the final census (2014) minus the total AGC stocks in the initial census (2003) divided by the elapsed time (i.e. 11 years) (Table 1).

**Statistical analysis**
*Disturbance and competitive thinning.* We used the size-dependent parameter of mortality (β) derived from the logistic regression to differentiate plots along a gradient of disturbance that ranges from sites strongly influenced by competitive thinning post internal disturbance (low β) to sites more influenced by active disturbances (high β). The size-dependent mortality parameter (β) was calculated as the probability of tree death (P) as a function of DBH according to logistic regression (logit(P) = a + β × DBH; if β < 0, smaller trees have a higher probability of dying due to competitive thinning during post internal disturbance, and if β > 0, larger trees have a greater probability of dying as expected under active disturbances[29]. The β parameter was also compared with the mean square diameter (Dq) of each plot. Dq was calculated by $Dq = \sqrt{\frac{\sum DBH_i^2}{n}}$; where DBH = diameter at breast height, and $n$ = the total number of stems in each plot. As a separate means of characterizing competitive thinning, we graphically evaluated the temporal changes in mean square diameter (Dq) and stem density. An increase in the number of individuals along with a decrease in Dq is expected in recently disturbed plots where the large trees have been lost and the recruitment of small stems is increasing; an increase in Dq but a decrease in the number of individuals is expected under competitive thinning[29].

*Latitudinal and elevational patterns of aboveground carbon stocks and dynamics.* We used generalized additive models (GAMs) to evaluate the pattern of change of AGC stocks and AGC dynamics (AGC net change, AGC productivity, and AGC mortality) along the latitudinal and elevational gradients. A One Way Analysis of Variance (ANOVA) was employed to evaluate differences in both AGC stocks and dynamics among countries.

*Drivers of aboveground carbon dynamics.* To select the most important explanatory variables of AGC dynamics along both latitudinal and elevational gradients in the 119 South American forest plots, we used Structural Equation Modelling (SEM)[26] to assess and visualize the likely multiple correlations between explanatory variables and their exogenous or latent influence on AGC dynamics. Overall, the SEM approach was employed to analyze the direct and indirect effect of climate (PCAs), symbiotic root association (SRA), phylogenetic diversity (PDz), directional compositional change (TR), initial aboveground carbon (AGC), and the size-discriminant β parameter associated to disturbance on AGC net change, AGC productivity, and AGC mortality. All variables were standardized before their inclusion in the SEM. The SEM included climate and SRA as exogenous variables. Climatic variables affected all variables in the model. In contrast, SRA only affected AGC1 and AGC dynamics. PDz was included as an explanatory variable of both AGC1 and dynamics. We controlled for the effect of climate and TR on the variation of PDz. AGC1 was affected by all explanatory variables, and affected both β and AGC dynamics. The β parameter directly influenced AGC dynamics, and was also affected by TR. We used a Satorra–Bentler scaled chi-square test statistic to determine whether the covariance matrix observed in our data significantly deviated from that predicted by the SEM[26]. Finally, we used estimates of standardized coefficients in each path of the model and R$^2$ for each endogenous variable to assess the importance of each set of explanatory variables for determining changes in the AGC dynamics along the latitudinal and elevation gradients in the Andean mountains of South America.

As a complementary analysis, we also used an information-theoretic (IT) approach[27,65]. The IT approach employs model-based inferences to generate a set of candidate models that represent competing hypotheses including different sets of explanatory variables. The IT model selection uses the relative Kullback-Leibler information to compute the difference in information loss between each model and reality. The relative Kullback-Leibler information is in turn assessed by the Akaike Information Criterion (AIC)[65], which identifies the best model as the one that minimizes the overall difference in information between the model and the ecological reality[66].

We used a natural model-averaging technique to select the most important explanatory variables of AGC net change, AGC productivity, and AGC mortality. The explanatory variables were associated with our hypotheses (Supplementary Table 1) on the relative importance of climate (PCAs), symbiotic root association (SRA), phylogenetic diversity (PDz), compositional change (TR), initial aboveground carbon (AGC1), and the size-discriminant β parameter as determinants of AGC dynamics. We assessed the nested effect of plots within countries as a means of differentiating geographic subregions, but it was not significant ($p > 0.05$). The set of models representing competing hypotheses were those models that had a net difference in AIC ≤ 4[65]. The natural model-averaging technique allows us to infer and predict AGC net change, AGC productivity, and AGC mortality based on a set of alternative models. The natural model-averaging technique calculates the average of each variable's parameter estimates over the models where the variable was selected[66,67]. The explanatory variables were previously standardized to have mean = 0 and standard deviation = 1, and then each parameter was standardized by the partial standard deviations[68]. The use of the partial standard deviations[68] aims to correct for the likely effect of multiple correlation among variables. The partial standard deviation (s*$_{xi}$) is calculated as follows:

$$S_{xi}^* = S_{xi}\sqrt{\left(VIF_{xi}^{-1}\right)} \times \sqrt{\left(\frac{(n-1)}{(n-p)}\right)}$$

where S$_{xi}$ is the standard deviation of the X$_i$ variable; VIF is the variance inflation factor; n is the number of observations; p the number of predictors in the model. The standardized coefficients (β$_i$*) are then transformed as β$_i$* = β$_i$ S*$_{xi}$, where β$_i$ is the unstandardized coefficient. All the information-theoretic (IT) analyses were performed with the *MuMIn* package in R.

We used both SEM and IT independently in a two-phase analysis. In the first phase, we assessed the importance of only the abiotic explanatory variables (PCA$_{temp}$1,2 + AGC1) in determining AGC dynamics. By doing so, we allow regional projections based on widely used and available abiotic information. In the second phase, we assessed the importance of both the abiotic and the biotic variables (TR, SRA, PDz, and β) in determining AGC dynamics. Note that we only used the temperature-derived PCA axes (PCA$_{temp}$1,2) as climatic explanatory variables in these analyses due to high collinearity with the PCA$_{prec}$1,2 axes (Supplementary Fig. 6, Supplementary Table 9).

## Data availability
The census data used to run the analyses are available as https://doi.org/10.5061/dryad.59zw3r26f.

## Code availability

The code employed to run the analysis will be available under request.

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

## Acknowledgements

A.D. received funding from the Dirección de Investigaciones de la Universidad Nacional de Colombia Sede Medellín (DIME). F.C. was funded by the Swiss Agency for Development and Cooperation (SDC), Grant/Award Number: PA0042-C011-0031 and by Universidad de las Américas Research Grant: FGE.FCC.19.15. S.G.C. was funded by COLCIENCIAS (Convocatoria 767). The Madidi Project has been supported by grants from the National Science Foundation (DEB-0101775, DEB-0743457 and DEB-1836353), the National Geographic Society (NGS 7754-04 and NGS 8047-06), The Living Earth Collaborative and various other organizations and individuals. K.J.F. was funded by an award by the US National Science Foundation (DEB LTREB 1754664). O.L.P. was supported by an ERC Advanced Grant 291585 ("T-FORCES"). P.K. was funded by the US National Science Foundation (DEB 1753621) and an Institute of International Education Fulbright Fellowship. We acknowledge the support provided by the Swiss Cooperation through the Programa Bosques Andinos conducted by CONDESAN who supported the workshop where the idea of this study was conceived as part of the core activities promoted by the Red de Bosques Andinos (RBA).

## Author contributions

A.D. and F.C. conceived the project. A.D., M.A.P., F.C., S.G-C., M.C., and P.K. built the general structure of the datasets. A.D., M.A.P., F.C., M.C., S.G.-C., P.K., and K.J.F. analyzed the data. A.D., F.C., S.G.-C., P.K., O.L.P., and K.J.F. wrote the paper. A.D., M.A.P., F.C., O.L.P., C.B., J.C., L.C., W.F.-R., A.F., R.G., J.H., M.I.L.-R., Y.M., A.M., L.M., J.A.M-V., J.A.M., O.O.-A., M.P., E.P., S.S., M.S., J.S.T., A.T.-V., and K.J.F. collected field data. All co-authors commented on and/or approved the manuscript.

## Competing interests

The authors declare no competing interests.
