## [Peer Review File · Nature Communications]

REVIEWER COMMENTS

Reviewer #1 (Remarks to the Author):

Nature Communications NCOMMS-20-35887-T

Andean forests as globally important carbon sinks and future carbon refuges

The authors present a study of carbon storage trends and develop models to explain factors such as climate, changes in species composition, and size dependent mortality. Their widely distributed plot network with repeat measurements allows them to examine trends both across the entire study area and (primarily through variations in climate) within regions. They found that carbon storage is increasing rapidly, and their models suggest several processes that appear to be most influential in the carbon dynamics such as initial carbon stock, size-dependent probability of mortality, and symbiotic root associations. In general relationships are weak (e.g., latitude and elevation of plot) to moderate (e.g., above ground carbon productivity).

The finding of the direction and size of carbon storage makes this study, in my opinion worth publication. The modest results in attempting to understand the causes of variations in that growth could be explained by ecology being messy, the wide range of conditions across their study area, or perhaps by a bias in their plots.

Line 93 is the first place where the authors discuss that the Andean forests are a mosaic of successional stages, but this is also brought up in a number of other places (e.g., line 252). The authors also state that the small size of the forest fragments that many of the plots are in suggest prior human disturbance. Elsewhere in the manuscript and figures there are hints that many of the plots represent relatively young forests where both rapid carbon uptake and competitive exclusion mortality would be expected (e.g., supplementary figure 8). It's well known that forests in different successional stages have different carbon and ecological dynamics. The hints that many to most of the plots represent early successional stages bring up multiple questions such as:

- How representative are the plots of successional stages prevalence in the Andean mountains? The authors could look at maps of successional stage or carbon to judge how representative their plots are.
-
- If many of the plots are early succession, then perhaps the models presented represent early successional ecological processes and relationships rather than processes uniquely associated with faster carbon uptake.
-

I have two specific suggestions:

- The authors show data that describes their plots in terms interpretable as successional stage. They could do a scatterplot of mean dbh vs stem count or mean dbh vs carbon storage per ha (actually, both would be interesting to see). It also would be interesting to see representative plots of lowland forests also shown since the authors set up this study as a contrast to lowland forest carbon storage and dynamics.
-
- If it appears that there are multiple successional stages, the authors should consider rerunning their models by different successional stage to see if they are reporting general or successional-stage specific results.

-

Many of the relationships the authors present between carbon dynamics and explanatory factors are weak. This may be because ecology, especially across such a wide range of conditions, is inherently messy or because they are applying models across inherently dissimilar successional stages.

If early successional forests are common in the area, then one would assume that there is a shifting mosaic of carbon uptake and loss that a larger scales results in a stable shifting pattern of carbon storage. The authors might want to comment on this and discuss whether their results indicate a preponderance of early succession plots in their study (perhaps due to recent human activity) or whether there is a fundamental shift in the ultimate carbon storage within this shifting mosaic.

The authors present many explicit and implicit hypotheses for possible drivers of carbon dynamics. I think that it would help the readers if they provided a table that states each hypothesis, the metric used to test each hypothesis, and the conclusion about each hypothesis. I found myself having to flip back and forth through the manuscript to keep the larger picture in mind. It would also help if this table provided the acronyms used in this study; I had trouble keeping those straight, too.

When Size-discrimination (β) is introduced in text and in each caption, the authors should indicate what higher values indicate.

I have relatively few specific comments; the manuscript is well organized and the data and results are clear.

Line 138: Additional information on the plots would be useful here. Are the plots in primary forests? Secondary forests? Plantations?

Line 166: The text should reflect which of these relationships were statistically significant.

Line 174: Could the authors state the direction of the correlations of nAGC with each of these factors?

Line 181: Knowing the types of forests in which the plots were located would help in interpreting the conclusion stated in this sentence. For example, if all plots were in long-established primary forests, then one would expect a steady state replenishment of smaller trees. However, if the plots are in secondary forests, especially younger ones, then competitive exclusion would be expected to be a major ecological process.

Line 256: This comment suggests chronic small scale disturbances across the Andes forests. If all other factors were held steady, then we would expect a steady state carbon storage and a steady state regeneration of smaller trees. However, if the increase of carbon storage is likely caused by the loss of smaller trees (allowing faster growth of the surviving larger trees), then what factor has led to the increase in small tree mortality?

Line 342: The authors are addressing the nature of the forest plots here. This is important information for the interpretation of results and should be summarized in the main body of the manuscript. The discussion should also include the impact of the nature of the conditions in the forests on the interpretation of the results.

Supplementary Fig. 4. – Figures should show the variance explained for each of the axes

Reviewer #2 (Remarks to the Author):

This study analyzes original data from 119 forest-monitoring plots across the Andes, showing that these forests are, on average, C sinks, and linking woody productivity, mortality, and net change in live biomass to environmental and biological factors. It is by far the largest study on this theme for Andean forests. The study provides novel insights into carbon regulation by Andean forests, which is extremely timely and relevant for climate change mitigation. The manuscript is generally very well-written.

I have a lot of relatively minor comments and questions about the analyses (see specific comments), and one bigger-picture comment: Overall, the large number of potential explanatory variables considered (most with novel abbreviations), the use of PCA analysis to characterize climate, and the complex relationships among them makes it time-consuming to fully understand the results. More fundamentally, inclusion of numerous biotic variables in the statistical models predicting prediction woody productivity, mortality, and net change in live biomass precludes predicting these variables at a regional level based on widely available data (land cover, elevation, climate, AGB to the extent that you trust satellite maps). As a result, the regional analysis simply relies on averages across elevational bands (which the statistical models show to be less influential than latitudinal variation). This is a notable disconnect. I'd prefer to see a model limited to abiotic drivers (maybe including AGB) that serves as the basis for the regional projection, and then the abiotic+biotic model that gives more insight into the underlying biological dynamics.

Overall, I think this an excellent and important study that warrants publication in a high-profile journal.

Specific comments:

Line 56-57- "almost everything we know about carbon regulation by forests comes from lowland ecosystems" – Please ensure that this statement is fair, perhaps reword to be more specific or slightly less strong. There have been some nice elevational gradient studies on C cycling in the Andes, although I don't remember offhand if they get at woody mortality and net carbon balance, as this study does.

Ref. 2 –Perhaps it's citation belongs at the end of the sentence ending line 74, rather than midway? I don't think it estimates total C sequestration.

Lines 147-148 (and throughout)- This is the first time I've seen these variable abbreviations. While this isn't wrong, and there isn't a single agreed-upon standard, it would be more intuitive to readers to use common abbreviations. For example, Δ AGB (or Δ AGC) instead of n_AGC; ANPPstem, ANPPwoody, or CWP (coarse woody productivity) instead of p_AGC; and Mwoody or CWM (coarse woody mortality) instead of m_AGC.

Lines 152-162- This paragraph introduces a lot of novel abbreviations (n=4, adding to those mentioned in previous comment), and their use throughout makes the paper more challenging to read. I recognize that word count is probably a challenge, but if possible it would help to use fewer novel abbreviations.

Lines 161, 180- In one of these places, please give a brief explanation of beta so that readers can understand the direction of the relationship without referring to the methods (e.g., Δ AGC was negatively associated with disproportionate mortality of large trees (high β)).

Lines 174-175- PCAtemp2 was associated MAINLY with the south-north climate gradient, but also correlated with elevation

Lines 181-184- I find the wording here strange. I wouldn't see small tree deaths as the cause AGC increases, but as a consequence of being outcompeted by growing larger trees.

Lines 192-193- Understanding this result requires having read the methods (see comment re lines 494-495).

Lines 195-196- This statement doesn't make sense to me. WHAT appeared to be largely associated with colder temperatures at higher elevations?

Paragraph starting line 221- Here, a very brief description of methods is needed to explain the area over which calculations were made.

Line 263- A forthcoming authoritative reference on CO2 is Walker et al., *New Phyt*, in press: <https://doi.org/10.1111/nph.16866>.

Analysis- it doesn't make sense (mechanistically) to fit a linear relationship between latitude and C variables when latitude spans the equator. I'd recommend either considering absolute latitude (preferred) or fitting a polynomial that would allow symmetry across the equator.

Analysis- I'm not convinced that presenting all the results excluding the subtropical forests is necessary. (I don't have an objection to presenting it, but think maybe it takes up space that may be better allocated elsewhere.)

Analysis- Mortality, particularly of large trees, is stochastic and best characterized by large plots. Do you obtain similar / cleaner results if analysis is limited to the larger plots in the network?

Line 351- "...the DBH of the second census was modified to match these extreme growth values⁴¹." – I don't understand what's meant here.

Lines 493-503- The statement on lines 493-494 seems in conflict with the following paragraph, and the order is also funny. PDz is mentioned on line 494 before it is defined in the next paragraph.

Lines 494-495- This statement is somewhat surprising/ counterintuitive (at least to me). I think it's important that (1) the point be made more prominently and earlier in the manuscript, as I think most readers would expect higher PDz near the equator, and (2) you add a sentence somewhere explaining why this is the case.

Line 562- Unless I'm missing something, the Dq metric here does not appear to ever come out as significant or be mentioned in the main text and figures. Can you drop it?

Line 564- why the superscript 2 ahead of the square root sign in the equation?

Line 572-573- country is an artificial distinction from a biological standpoint (although geographical clustering makes it reasonable to include this)

Fig. 1- align maps, increase legend font size

Fig. 2- I'd probably drop the 2nd column of panels (elevation), given that there are no significant

trends.

Fig. 3- differences in arrow colors look pretty subtle on my screen; please increase the contrast between arrow colors. Figure would also be more intuitive if variables were listed as titles to each plot, and at a minimum caption needs to link the panels (a-c) to the variables in a first, brief figure caption summary.

Figures- I'd love to see a map figure presenting biomass estimates across the Andes and corresponding to the regional estimates presented in the last paragraph of the results. One option would be to put this in Fig. 1.

SI figures- It seems these should be ordered as referenced in the text, so methods figures on height allometries come later.

Table S4- Do the AGC rate variables ever appear in the main article? (I don't see them, but perhaps I'm missing something.) If not, they should just be deleted. I also find it to be a funny metric—tricky to interpret, and I'm not sure of its value.

REVIEWER COMMENTS

Reviewer #1 (Remarks to the Author):

Nature Communications NCOMMS-20-35887-T

Andean forests as globally important carbon sinks and future carbon refuges

R1: The authors present a study of carbon storage trends and develop models to explain factors such as climate, changes in species composition, and size dependent mortality. Their widely distributed plot network with repeat measurements allows them to examine trends both across the entire study area and (primarily through variations in climate) within regions. They found that carbon storage is increasing rapidly, and their models suggest several processes that appear to be most influential in the carbon dynamics such as initial carbon stock, size-dependent probability of mortality, and symbiotic root associations. In general relationships are weak (e.g., latitude and elevation of plot) to moderate (e.g., above ground carbon productivity).

A: In large-scale studies in tropical forests, but particularly in montane forests, the correlation between AGC dynamics and environmental variables is usually expected to be “weak” due to the high heterogeneity both in terms of forest type and in environmental variables. Actually, the magnitude of the associations between AGC dynamics and environmental variables reported in this study are relatively large in the context of large-scale tropical forest analyses (cf, for example, the strength of relationships reported here with those in the recent tropical large-scale tropical lowland forest analyses in Fig. 3 of Hubau et al. 2020).

R1: The finding of the direction and size of carbon storage makes this study, in my opinion worth publication. The modest results in attempting to understand the causes of variations in that growth could be explained by ecology being messy, the wide range of conditions across their study area, or perhaps by a bias in their plots.

A: We appreciate the overall comments about the value of our study. We don't agree, however, that there is a ‘bias’ in the plots. To resolve any misunderstanding, which we

believe was likely due to our clear lack of description of the plots in the main text, we have provided new wording to better explain the general conditions of our plots, which are neither secondary nor plantations. We think this new description (L 128-134) will help address some of the comments and interpretations of R1.

R1: Line 93 is the first place the first place where the authors discuss that the Andean forests are a mosaic of successional stages, but this is also brought up in a number of other places (e.g., line 252). The authors also state that the small size of the forest fragments that many of the plots are in suggest prior human disturbance. Elsewhere in the manuscript and figures there are hints that many of the plots represent relatively young forests where both rapid carbon uptake and competitive exclusion mortality would be expected (e.g., supplementary figure 8). It's well known that forests in different successional stages have different carbon and ecological dynamics. The hints that many to most of the plots represent early successional stages bring up multiple questions such as:

A: Here we aimed to acknowledge the likely historical anthropogenic influence on Andean forests, as this is an important distinguishing feature of these forests compared to those in Amazon lowlands. Ignoring the influence of pre- and post-hispanic colonization human on shaping forest structure in the Andes would represent a real bias in a study at this large scale, but, germane to R1's comments, we didn't include secondary forests in our study. This point is clarified in the new version (L 128-134), where we have emphasized the likely effect of disturbance (either natural or human-induced) on shaping the trend of recovery in some of our plots. We think the new description of the sites will help readers to better understand the underlying nature of the forests we studied.

R1: How representative are the plots of successional stages prevalence in the Andean mountains? The authors could look at maps of successional stage or carbon to judge how representative their plots are.

A: As noted above, we make clear in the new main text that we do not include secondary forests. The use of the size-dependent parameter of mortality (β), aimed to differentiate the phase of development of each plot (see lines 159-164). Unfortunately, we are not aware of any reliable map that could describe specific stages of forest development in the Andean region at the 1-ha scale. Indeed, such a map would be hard to generate without also accounting for long-term plot-based records, which are important to validate any attempt to define forest developmental stages.

R1: If many of the plots are early succession, then perhaps the models presented represent early successional ecological processes and relationships rather than processes uniquely associated with faster carbon uptake.

A: Again, as pointed out above, we have included a better description of the plots in the main text where we clarify the lack of inclusion of secondary forests. Furthermore, we used the quartiles of the size-dependent parameter of mortality (β) to differentiate plots influenced by post-disturbance effects (self-thinning), and analyzed their influence on the overall AGC net change. We hope this new analysis will allay the concern made here by R1.

R1: I have two specific suggestions:

R1: • The authors show data that describes their plots in terms interpretable as successional stage. They could do a scatterplot of mean dbh vs stem count or mean dbh vs carbon storage per ha (actually, both would be interesting to see).

A: While the suggested scatterplots are shown below (Fig. R1), we don't think they provide any useful insight into our study results. First, we guess R1 means mean square diameter and not mean dbh, which is not an appropriate metric. More importantly, along elevational gradients, where AGC stocks decline with elevation (Fig. S10 in the new version), the typical declining trend of stem density along with the increase of tree size can't be interpreted as an outcome of self-thinning. Thus, the relationships depicted in these scatterplots are of the form expected in any heterogeneous collection of forest samples including a wide climatic and geographic range (as in our analysis). This is why we preferred to show the trend of change between initial and final censuses of each plot (Fig. S7B in the new version) rather than independently for each census, as asked by R1. In the new version, we think that new Figures S7 and S11 cover the suggestions made by R1 for interested readers.

Figure R1. Relationships between stem density (ha⁻¹) and aboveground carbon stocks of the initial (I) and final (F) censuses and the respective mean square diameter (Dq) in the 119 plots located in the subtropical and tropical Andes.

R1: It also would be interesting to see representative plots of lowland forests also shown since the authors set up this study as a contrast to lowland forest carbon storage and dynamics.

A: In the new version, we have divided the β parameter in quartiles and used them as a way to classify the dynamical stage of the plots (L 212-218). Then, we assessed and discussed the effect of ruling out the plots with values of β below the first quartile (0.25), which are assumed to be those with a higher mortality of small stems due to post-disturbance tree growth recovery. We believe this analysis is most in line with what R1 is seeking here. The AGC net change of plots with small or no influence of post-disturbance (between β quartiles 0.25 and 1.0), were then compared with the AGC net change reported in the Amazon lowlands (L 271-273).

R1:• If it appears that there are multiple successional stages, the authors should consider rerunning their models by different successional stage to see if they are reporting general or successional-stage specific results.

A: We reiterate our apology for any lack of clarity in the original text. The use of the β parameter, which is a continuous variable, as an explanatory variable, allows us to keep the overall degrees of freedom intact instead of splitting the developmental plot stages into smaller categories. We believed that the analysis explained in the previous answer fully addresses the main query made here by R1.

R1: Many of the relationships the authors present between carbon dynamics and explanatory factors are weak. This may be because ecology, especially across such a wide range of conditions, is inherently messy or because they are applying models across inherently dissimilar successional stages.

A: We agree with the first point raised by R1: ecology is inherently messy. See below for an answer to the first comment made by R2, which is very similar to this one.

R1: If early successional forests are common in the area, then one would assume that there is a shifting mosaic of carbon uptake and loss that a larger scales results in a stable shifting pattern of carbon storage. The authors might want to comment on this and discuss whether their results indicate a preponderance of early succession plots in their study (perhaps due to recent human activity) or whether there is a fundamental shift in the ultimate carbon storage within this shifting mosaic.

A: To address this suggestion, we have divided the amount of carbon gains between self-thinning and “ultimate carbon storage” due to other factors, such as those associated with global warming and climate change (L 271-276).

R1: The authors present many explicit and implicit hypotheses for possible drivers of

carbon dynamics. I think that it would help the readers if they provided a table that states each hypothesis, the metric used to test each hypothesis, and the conclusion about each hypothesis. I found myself having to flip back and forth through the manuscript to keep the larger picture in mind. It would also help if this table provided the acronyms used in this study; I had trouble keeping those straight, too.

A: Thank you for this suggestion. We included a new table in the supplementary information (Table S4) in which we addressed the comment made by R1. Since we received a similar comment from R2 about the description of the explanatory variables, we believe the inclusion of the new supplementary table will help the reader to clarify the meaning of each abbreviation and hypothesis stated.

R1: When Size-discrimination (β) is introduced in text and in each caption, the authors should indicate what higher values indicate.

A: On lines 159 and 163, we have included an explicit description of what the β values mean to avoid confusion.

R1: I have relatively few specific comments; the manuscript is well organized and the data and results are clear.

R1: Line 138: Additional information on the plots would be useful here. Are the plots in primary forests? Secondary forests? Plantations?

A: See the new wording on lines 128 and 134 that clarify the type of forests sampled in this study.

R1: Line 166: The text should reflect which of these relationships were statistically significant.

A: Unfortunately, we do not understand this comment. Here, we are simply presenting the means and their associated uncertainty (SD or SE), which does not need any test of significance.

R1: Line 174: Could the authors state the direction of the correlations of nAGC with each of these factors?

A: On lines 207 and 211, we have included the direction of the correlations.

R1: Line 181: Knowing the types of forests in which the plots were located would help in interpreting the conclusion stated in this sentence. For example, if all plots were in long-established primary forests, then one would expect a steady state replenishment of smaller trees. However, if the plots are in secondary forests, especially younger ones, then competitive exclusion would be expected to be a major ecological process.

A: As noted above, we have clarified that we are not including secondary forests. In this new version, between lines 128-134, a brief but more complete description of the features

of the plots is included.

R1: Line 256: This comment suggests chronic small-scale disturbances across the Andes forests. If all other factors were held steady, then we would expect a steady state carbon storage and a steady state regeneration of smaller trees. However, if the increase of carbon storage is likely caused by the loss of smaller trees (allowing faster growth of the surviving larger trees), then what factor has led to the increase in small tree mortality?

A: We hypothesized here about the influence of global warming and the CO₂ and N increase (fertilization effects) as likely causes of about 50% of the C gains. We used the β parameter to differentiate the developmental stage of plots (see L 273-276).

R1: Line 342: The authors are addressing the nature of the forest plots here. This is important information for the interpretation of results and should be summarized in the main body of the manuscript. The discussion should also include the impact of the nature of the conditions in the forests on the interpretation of the results.

A: We have now included a detailed description of the plots as part of the main text to avoid confusions about the nature of the plots. In the discussion, as explained above, the influence of the developmental stage of the plots on the AGC net change was also addressed in several sections.

R1: Supplementary Fig. 4. – Figures should show the variance explained for each of the axes

A: The percentage of variance explained by each PCA axis is now included in the figure.

Reviewer #2 (Remarks to the Author):

R2: This study analyzes original data from 119 forest-monitoring plots across the Andes, showing that these forests are, on average, C sinks, and linking woody productivity, mortality, and net change in live biomass to environmental and biological factors. It is by far the largest study on this theme for Andean forests. The study provides novel insights into carbon regulation by Andean forests, which is extremely timely and relevant for climate change mitigation. The manuscript is generally very well-written.

A: We sincerely appreciate all the comments received by R2 and believe they have helped us to significantly improve the overall quality of the manuscript.

R2: I have a lot of relatively minor comments and questions about the analyses (see specific comments), and one bigger-picture comment: Overall, the large number of potential explanatory variables considered (most with novel abbreviations), the use of PCA analysis to characterize climate, and the complex relationships among them makes it time-consuming to fully understand the results. More fundamentally, inclusion of numerous biotic variables in the statistical models predicting prediction woody productivity, mortality, and net change in live biomass precludes predicting these variables at a regional

level based on widely available data (land cover, elevation, climate, AGB to the extent that you trust satellite maps). As a result, the regional analysis simply relies on averages across elevational bands (which the statistical models show to be less influential than latitudinal variation). This is a notable disconnect. I'd prefer to see a model limited to abiotic drivers (maybe including AGB) that serves as the basis for the regional projection, and then the abiotic+biotic model that gives more insight into the underlying biological dynamics.

A: Thank you very much for these comments. We have followed R2's suggestions and included a new section that differentiates between abiotic + AGC 1 and abiotic + biotic explanatory variables of the AGC dynamics. Note we also address R2's suggestion and left the AGC stocks of the census along with the abiotic variables. We provide a full description of the modified methods between lines 147 to 164 and throughout the results section. One important issue is that the abiotic variables remaining in the models were always significant also in the models including both abiotic and biotic variables, which facilitates their discussion.

R2: Overall, I think this an excellent and important study that warrants publication in a high-profile journal.

A: Thank you for the very positive evaluation.

Specific comments:

R2: Line 56-57- "almost everything we know about carbon regulation by forests comes from lowland ecosystems" – Please ensure that this statement is fair, perhaps reword to be more specific or slightly less strong. There have been some nice elevational gradient studies on C cycling in the Andes, although I don't remember offhand if they get at woody mortality and net carbon balance, as this study does.

A: We have modified the wording. Now we say: "*However, estimates of tropical C uptake are largely based on studies of lowland ecosystems*"

R2: Ref. 2 –Perhaps it's citation belongs at the end of the sentence ending line 74, rather than midway? I don't think it estimates total C sequestration.

A: Fixed.

Lines 147-148 (and throughout)- This is the first time I've seen these variable abbreviations. While this isn't wrong, and there isn't a single agreed-upon standard, it would be more intuitive to readers to use common abbreviations. For example, Δ AGB (or Δ AGC) instead of n_AGc; ANPPstem, ANPPwoody, or CWP (coarse woody productivity) instead of p_AGc; and Mwoody or CWM (coarse woody mortality) instead of m_AGc.

A: The name of the AGC associated variables were changed simply to: AGC net change, AGC productivity, and AGC mortality. A new supplementary table (Table S4) in which we show the acronyms and definitions was also included (as suggested by R1).

R2: Lines 152-162- This paragraph introduces a lot of novel abbreviations (n=4, adding to those mentioned in previous comment), and their use throughout makes the paper more challenging to read. I recognize that word count is probably a challenge, but if possible it would help to use fewer novel abbreviations.

A: The definition of the biotic explanatory variables was now introduced in the previous paragraphs, and also clearly described between lines 153 and 164. We agree with R2 that there was a need of moving some text from Materials and Methods to the main text to help understanding better the main rationale of the hypotheses and models.

R2: Lines 161, 180- In one of these places, please give a brief explanation of beta so that readers can understand the direction of the relationship without referring to the methods (e.g., Δ AGC was negatively associated with disproportionate mortality of large trees (high β)).

A: On lines 159 and 163, we have included a clear description of the use and interpretation of the size-dependent parameter of mortality (β) as part of the main text.

R2: Lines 174-175- PCAtemp2 was associated MAINLY with the south-north climate gradient, but also correlated with elevation

A: Perhaps this is a misunderstanding by R2. PCAtemp2 is not associated with elevation (Fig. S6D), but it is with precipitation. In this new version, we first include a new table showing the effect of colinearity between PCA_{temp} and PCA_{prec} in the SEM models (Table S5), and, second, a brief paragraph in the discussion about its effect (lines 207-211).

R2: Lines 181-184- I find the wording here strange. I wouldn't see small tree deaths as the cause AGC increases, but as a consequence of being outcompeted by growing larger trees.

A: The wording was modified as follows: “*The strong effect of β on the AGC net change suggests that much of the recorded increase in AGC net change in tropical and subtropical Andean forests over the past two decades has been due to post-disturbance growth of larger trees in plots where competitive thinning enhanced the probability of mortality for small trees (Supplementary Fig. 7).*” (L 214-215)

R1: Lines 192-193- Understanding this result requires having read the methods (see comment re lines 494-495).

A: This text was modified as: “*Collectively, this indicates that Andean forest AGC productivity is higher near the equator in plots with higher initial AGC stocks, greater proportional abundance of AM fungal associations, and lower phylogenetic diversity (Fig. 3B; Supplementary Table 7).*” (L 222-225).

R2: Lines 195-196- This statement doesn't make sense to me. WHAT appeared to be largely associated with colder temperatures at higher elevations?

A: Deleted.

R2: Paragraph starting line 221- Here, a very brief description of methods is needed to explain the area over which calculations were made.

A: We included in the main text a brief description of the methods applied to assess the overall C balance in Andean forests (lines 175-190). See also Materials and Methods.

R2: Line 263- A forthcoming authoritative reference on CO₂ is Walker et al., New Phyt, in press: <https://doi.org/10.1111/nph.16866>.

A: We included the suggested reference.

R2: Analysis- it doesn't make sense (mechanistically) to fit a linear relationship between latitude and C variables when latitude spans the equator. I'd recommend either considering absolute latitude (preferred) or fitting a polynomial that would allow symmetry across the equator.

A: Although this comment has an intuitive reasoning, here we sought to show the full geographical-spatial south-to-north pattern instead of focusing on specific functional relationships with latitude. Note that if we use the absolute latitudinal values instead of those employed here, the trend would show a (distorted) humped pattern with a trend decreasing towards the equator, likely shaped by local features rather than by issues associated with their latitudinal position. Given this, we have not made either of the changes suggested by R2 here.

R2: Analysis- I'm not convinced that presenting all the results excluding the subtropical forests is necessary. (I don't have an objection to presenting it, but think maybe it takes up space that may be better allocated elsewhere.)

A: We followed your suggestion and removed the issues associated with results that excluded the subtropical forests. Thank you!

R2: Analysis- Mortality, particularly of large trees, is stochastic and best characterized by large plots. Do you obtain similar / cleaner results if analysis is limited to the larger plots in the network?

A: Ninety-four (94) out of the 119 plots are ≥ 1 -ha. We have made this distinction clear in the main text. Excluding the plots smaller than 1-ha would completely exclude the data from Ecuador, which we think is an important contribution to our study. Therefore, employing the same data that will be attached as part of our study, we used the IT approach (perhaps the most powerful method to identify significant explanatory variables) to assess the influence of ruling out all plots ≤ 1 -ha on determining the drivers of AGC mortality. As you will see, the changes can be considered as minor. Overall, the thermophilization rate (TR) wasn't significant in this case (in green), but the remaining significant variables were the same as previously identified as significant (in red) by the IT method employing the

119 plots (Table S7). Although we can't discard the possibility that 1-ha plots are still too small in size to detect some patterning in tree mortality in Andean forests, the lack of very large plots prohibits comparison of our findings with other lowland studies where there was no obvious effect of plot size on AGB dynamics (e.g. supplementary information in Lewis et al. 2009). Therefore, we have only included some discussion at the end of the second paragraph of the discussion (L 279-282) in which we acknowledge the need for future studies to use larger sample sizes in general to better differentiate further the main drivers of biomass dynamics in complex, tropical mountain forest ecosystems.

	Variable Parameter	MAE prom	MAE SE	P	RVI	No. Model
AGB mortality (Mg y ⁻¹ ha ⁻¹)	Intercept	0.000	0.000			
	AGC1	0.433	0.073	0.000	1.00	14
	PCAtemp1	0.143	0.074	0.057	0.28	5
	PCAtemp2	0.064	0.090	0.477	0.36	7
	TR	0.119	0.073	0.108	0.81	10
	PDz	-0.167	0.075	0.028	0.84	11
	SRA	0.096	0.079	0.226	0.65	9
	Beta	0.401	0.076	0.000	1.00	14

Table R1. Information theoretic (IT) modeling employing model-based inference to generate a set of candidate models that represent competing hypotheses build up by different sets of explanatory variables that explain aboveground carbon (AGC) mortality based on 94 plots (≥ 1 -ha). The competing hypotheses were represented by climate (PCAtemp1 and PCAtemp2), symbiotic root associations (SRA = $\ln(\text{AM}/\text{EcM})$), the thermophilization rate (TR; °C y⁻¹), the initial aboveground carbon stock in each plot (AGC1; C Mg ha⁻¹), and the standardized effect size of the phylogenetic diversity (PDz). MAE prom: model-averaged coefficient estimate. MAE SE: unconditional standard error. P: probability. RVI: relative variable importance. No. Model: Number of models that include the variable. Values in bold shows significant variables.

R2: Line 351- "...the DBH of the second census was modified to match these extreme growth values⁴¹." – I don't understand what's meant here.

A: The text was modified as follows: "*In cases where the recorded DBH growth of the second census was less than -0.1 cm y⁻¹ or greater than 7.5 cm y⁻¹, the DBH of the second census was augmented/reduced in order to match these minimum/maximum values⁴⁵.*" We hope it is clearer now.

R2: Lines 493-503- The statement on lines 493-494 seems in conflict with the following paragraph, and the order is also funny. PDz is mentioned on line 494 before it is defined in the next paragraph.

A: Lines 493-495 in the former version were deleted.

R2: Lines 494-495- This statement is somewhat surprising/ counterintuitive (at least to me). I think it's important that (1) the point be made more prominently and earlier in the manuscript, as I think most readers would expect higher PDz near the equator, and (2) you add a sentence somewhere explaining why this is the case.

A: On lines 503 – 508 we modified the text as follows: “SR and PDz were negatively correlated ($r = -0.59$, $p < 0.001$). This negative correlation is mainly due to the increased mixture of temperate-affiliated and tropical-affiliated species, both southwards and at higher elevations in the tropics^{57,58}. However, SR was never a significant explanatory variable in the models due to the greater relative importance of climate variables, and we only used PDz in the models as the surrogate of complementarity effects.”

R2: Line 562- Unless I'm missing something, the Dq metric here does not appear to ever come out as significant or be mentioned in the main text and figures. Can you drop it?

A: We believe that R2 may have missed a couple of figures and text where Dq was presented and used. In the Supplementary Figures 7 and 11, Dq was presented and actually, the trend of the β parameter is interpreted in relation to Dq.

R2: Line 564- why the superscript 2 ahead of the square root sign in the equation?

A: Removed.

R2: Line 572-573- country is an artificial distinction from a biological standpoint (although geographical clustering makes it reasonable to include this).

A: We agree that countries are not biogeographic entities, but still the best way to describe the regions. As such, we have decided to leave the text as it was.

R2: Fig. 1- align maps, increase legend font size

A: Fig. 1 has been completely re-done.

R2: Fig. 2- I'd probably drop the 2nd column of panels (elevation), given that there are no significant trends.

A: In this case, we disagree with R2. Specifically, we think that the lack of correlation between AGC dynamics with elevation at the scale of this study is itself an important result and likely to be of interest to readers.

R2: Fig. 3- differences in arrow colors look pretty subtle on my screen; please increase the contrast between arrow colors. Figure would also be more intuitive if variables were listed as titles to each plot, and at a minimum caption needs to link the panels (a-c) to the variables in a first, brief figure caption summary.

A: Thank you. We have edited Figure 3 accordingly.

R2: Figures- I'd love to see a map figure presenting biomass estimates across the Andes and corresponding to the regional estimates presented in the last paragraph of the results. One option would be to put this in Fig. 1.

A: In Figure 1 we have included an AGC net change scale to illustrate the overall regional trends presented in this study.

R2: SI figures- It seems these should be ordered as referenced in the text, so methods figures on height allometries come later.

A: They are cited in the main text right before the other figures (Line 137).

R2: Table S4- Do the AGC rate variables ever appear in the main article? (I don't see them, but perhaps I'm missing something.) If not, they should just be deleted. I also find it to be a funny metric—tricky to interpret, and I'm not sure of it's value.

A: The relative values were removed from the former Table S4 (now Table S6).

REVIEWERS' COMMENTS

Reviewer #1 (Remarks to the Author):

Nature Communications NCOMMS-20-35887-T

Andean forests as globally important carbon sinks and future carbon refuges

I believe that the authors have done a good job of clarifying my main concern – the nature of the forests studied – and in responding to my and the other reviewer's comments. I recommend that the manuscript be accepted with the authors possibly responding to my few minor suggestions.

This study makes important contributions to both understanding the size of the carbon sink and to understanding the forest dynamics and abiotic factors that influence the dynamics of carbon change.

First, the addition of lines 129 to 135 make it clear that the authors studied only primary forests. I suggest that they consider making it clearer to readers from the beginning that this is a study of primary forests. They could, for example add the word 'primary' in a few key places: Title: You might consider changing the title to: "*Primary* Andean forests as...". Similarly, line 99, "The AGC stocks and productivity of *primary* Andean forests...", and line 119, "...AGC dynamics in *primary* Andean forests."

Also, for readers (such as myself) who do not work in the Andes region, the authors might briefly state how extensive primary forests are across the range of their study. If they are a small proportion, then the implications of their study of these forests as a potential carbon sink are correspondingly small, and vice versa, with implications for global carbon dynamics. Even if the proportion of these forests is small, however, this is an important study elucidating the carbon dynamics of primary forests and compliments similar studies in primary forests in other regions.

This study uses many variables, all of which have their own acronyms. I was repeatedly looking to the caption of Figures 2 and 3 to remind myself of what the primary ones were. Readers will be able to digest this study more quickly if the authors include a table that defined the acronyms in one place.

I thank the authors for their new Supplementary Table 4; I found it useful in digesting their results. They might want to add a row considering size-dependent mortality since that proved to be an important driver of above ground carbon dynamics. I also encourage the authors to consider moving the table to the main body of the manuscript if the journal's space limitations allow. I believe that readers will be able to much more quickly digest the key messages of the manuscript if this table isn't buried in the supplementary material.

Dr. Van R. Kane
University of Washington

Reviewer #2 (Remarks to the Author):

I am satisfied with the responses to my previous comments and find the manuscript greatly improved. I have just a few very minor comments:

- line 201: perhaps note here "detailed below". On the first read I felt that insufficient information was given here

- line 262- you might highlight the fact that this is what the IPCC is using -- not just any estimate!

- Line 276- it would make more sense to replace "competitive thinning" with something like "succession" or "recovery from past disturbance"

signed,

Kristina Anderson-Teixeira